# Lip-Reading Advancements: A 3D Convolutional Neural Network/Long Short-Term Memory Fusion for Precise Word Recognition

**Themis Exarchos** 1,*, **Georgios N. Dimitrakopoulos** 1, **Aristidis G. Vrahatis** 1, **Georgios Chrysovitsiotis** 2, **Zoi Zachou** 2 and **Efthymios Kyrodimos** 2

1   Department of Informatics, Ionian University, 49100 Corfu, Greece; dimitrakopoulos@ionio.gr (G.N.D.); aris.vrahatis@ionio.gr (A.G.V.)
2   1st Otorhinolaryngology Department, National and Kapodistrian University of Athens, 11527 Athens, Greece; chrysovi@gmail.com (G.C.); zoizachou@gmail.com (Z.Z.); timkirodimos@hotmail.com (E.K.)
*   Correspondence: exarchos@ionio.gr

**Abstract:** Lip reading, the art of deciphering spoken words from the visual cues of lip movements, has garnered significant interest for its potential applications in diverse fields, including assistive technologies, human–computer interaction, and security systems. With the rapid advancements in technology and the increasing emphasis on non-verbal communication methods, the significance of lip reading has expanded beyond its traditional boundaries. These technological advancements have led to the generation of large-scale and complex datasets, necessitating the use of cutting-edge deep learning tools that are adept at handling such intricacies. In this study, we propose an innovative approach combining 3D Convolutional Neural Networks (CNNs) and Long Short-Term Memory (LSTM) networks to tackle the challenging task of word recognition from lip movements. Our research leverages a meticulously curated dataset, named *MobLip*, encompassing various speech patterns, speakers, and environmental conditions. The synergy between the spatial information extracted by 3D CNNs and the temporal dynamics captured by LSTMs yields impressive results, achieving an accuracy rate of up to 87.5%, showcasing robustness to lighting variations and speaker diversity. Comparative experiments demonstrate our model's superiority over existing lip-reading approaches, underlining its potential for real-world deployment. Furthermore, we discuss ethical considerations and propose avenues for future research, such as multimodal integration with audio data and expanded language support. In conclusion, our 3D CNN-LSTM architecture presents a promising solution to the complex problem of word recognition from lip movements, contributing to the advancement of communication technology and opening doors to innovative applications in an increasingly visual world.

**Keywords:** lip reading; word recognition; deep learning

## 1. Introduction

The visual signals encoded within lip movements have long fascinated researchers, linguists, and technologists in the field of human communication. The ability to decode spoken language from the intricate movements of the lips has transformative potential in a variety of domains, including enhancing accessibility for people with speech and hearing impairments and furthering human–computer interaction through unobtrusive communication. Rooted in the observation that lip movements carry valuable information about spoken language, the field has evolved from its historical foundations to embrace modern computational approaches [1].

Lip reading as a technology holds immense potential for transformative applications in various medical contexts. One significant application lies in the field of assistive technologies, where it can serve as a valuable tool for individuals with hearing impairments or with

vocal cord lesions [2]. By accurately interpreting spoken words through lip movements, this technology can enhance communication and accessibility for these people, contributing to an improved quality of life. Moreover, the integration of lip-reading technology into healthcare settings can have implications for patient–doctor communication. In scenarios where traditional communication channels may be limited, such as during medical procedures or in noisy environments, the lip-reading task can offer an additional means of understanding patient needs and concerns. This can foster better doctor–patient interactions and contribute to more effective healthcare delivery. Finally, lip reading has the potential to play a role in health monitoring and diagnostics. Changes in speech patterns or lip movements may be indicative of certain health conditions, and the integration of lip-reading technology into diagnostic tools could provide valuable insights for healthcare professionals. Beyond healthcare, such models can also be used for biometric identification, enhancing security [3]. Finally, they can be used in novel virtual reality applications, such as avatar creation, gaming, and interaction with computers/robots, or even in film making, since there already exist applications that generate realistic face animations from images based on audio [4].

In parallel, in the machine learning field, several advanced algorithms based on neural networks have been proposed to solve complex tasks in biomedical image analysis. For example, the Complementary Adversarial Network-driven Surface Defect Detection (CASDD) framework [5] was proposed to address the intrinsic challenges of intra-class differences. This framework integrates encoding–decoding segmentation, dilated convolutional layers, and a complementary discriminator mechanism. The Fusion Attention Block Network (FABNet) [6] introduced a model transfer method grounded in clinical experience and sample analysis. Additionally, an end-to-end Depth Domain Adaptive Network (DDANet) was proposed to address challenges related to interpretability in laryngeal tumor grading [7]. By integrating prior-experience-guided attention and integration gradient Class Activation Mapping (CAM), this approach achieved elevated grading accuracy and interpretability, bridging the gap between computer vision-based and pathology-based diagnoses. Based on a Transformer component, the Focal Loss-Swin-Transformer Network (FL-STNet) model was introduced for lung adenocarcinoma classification [8], which exhibited efficacy in capturing both the overall tissue structure and local details. In a parallel context, an adaptive model fusion strategy was created for breast cancer tumor grading, amalgamating Vision Transformer (ViT) and Convolutional Neural Network (CNN) blocks with integrated attention [9]. This approach attained an accuracy of 95.14%, surpassing ViT-B/16 and FABNet. Finally, Omar et al. [10] focused on optimizing epileptic seizure recognition using deep learning models. Various architectures were explored, such as Conv1D, Long Short-Term Memory (LSTM), bidirectional LSTM (BiLSTM), and Gated Recurrent Units (GRUs). The study identified the Conv1D-LSTM architecture, augmented with dropout layers, as more effective, while the impact of feature scaling, principal component analysis (PCA), and feature selection methods was highlighted. In the realm of facial recognition, Taha et al. [11] introduced a novel approach to recognize faces with masks, integrating mask detection, landmark detection, and oval face detection. This was performed using robust principal component analysis (RPCA) and a pretrained ssd-MobileNetV2 model for mask detection, and the features were optimized with the Gazelle Optimization Algorithm (GOA). This approach can be used in diverse applications, including security systems, access control, and public health measures. Collectively, these frameworks showcased innovative solutions tailored to address specific challenges in real-world applications of biomedical image processing.

Toward visual speech analysis, initial efforts included approaches mainly based on Hidden Markov Models [12,13]. However, during the last few years, the integration of deep learning models, marked by their ability to discern complex patterns within data, has propelled lip reading into a new era of accuracy and applicability [2,14]. In addition, several databases holding audiovisual data have been created [15], although predominantly for English and a few other widely spoken languages.

This paper embarks on a journey into the enchanting domain of lip reading, where the fusion of advanced algorithms and computer vision techniques elevates the subtleties of visual speech to an extraordinary level of comprehension. Through this exploration, we embrace the fusion of human perception and computational innovation, forging a path that converges human intuition with the transformative capabilities of technology. Motivated by the lack of data for the Greek language, we created the "MobLip" dataset, containing lip frames from 30 subjects, including sentences with several of the most common Greek words. Then, in order to maximize the accuracy of word recognition, we implemented and compared various different deep neural network architectures.

## 2. Related Work

Lip reading was revolutionized by the work of Assael et al. [16], who introduced LipNet, an innovative end-to-end model designed specifically for sentence-level comprehension. This study delves deeply into the architectural complexities of LipNet, emphasizing its innovative sequence-to-sequence framework that directly translates lip image sequences into coherent word sequences. The authors positioned LipNet as an innovative solution capable of deciphering the complexities of spoken language through visual signals by addressing the multifaceted difficulties of lip reading. The heart of the LipNet architecture was the sequence-to-sequence framework, which seamlessly integrated CNN and LSTM networks. This dynamic fusion empowered the model to automatically learn the intricate spatiotemporal relationships inherent in lip movements across time, creating a unified representation of lip gestures that directly informs the sentence-level interpretation. As a result, LipNet exhibited a remarkable capacity to capture both fine-grained visual details and temporal patterns, thus aligning with the complex nature of spoken language. To validate the efficacy of their creation, Assael et al. [16] conducted extensive experiments on a large-scale lip-reading dataset, evaluating LipNet's performance against existing benchmarks. On the GRID corpus, LipNet yielded a 95.2% accuracy, highlighting the model's ability to correctly predict words and sentences solely from the sequences of lip images. The results unveiled LipNet's remarkable ability to outperform traditional methods and excel at the challenging task of sentence-level lip reading. Notably, the model demonstrated proficiency across various accents, languages, and speaking styles, underlining its versatility in tackling the complexities of real-world lip reading.

Furthermore, three different architectures were proposed for lip reading sentences using the BBC LRS2 dataset by Afouras et al. [17]. The front-end of all three systems was composed of a 3D-CNN and a ResNet. The initial architecture used an external language model to aid with decoding, and its back-end comprised three stacked bidirectional LSTMs trained with a Connectionist Temporal Classification (CTC) loss. Based on the standard model presented in [18], the second system employed an attention-based transformer with an encoder–decoder architecture. In every evaluation scenario, the transformer model outperformed the bidirectional LSTM model, achieving 50% in terms of word accuracy. It was also found that the transformer model was superior to the bidirectional LSTM model in the task of generating longer sequences (those with more than 80 frames). Moreover, as the bidirectional LSTM model relied on the CTC's premise of conditionally independent time-step outputs, it was unable to learn long-term, nonlinear dependencies or model complicated grammatical rules.

Yang et al. [19] proposed an architecture for lip reading Chinese words from the LRW1000 dataset named the D3D model. It consisted of a front-end with a spatiotemporal CNN whose topology was similar to that of DenseNet, with phases of convolution, batch normalization, and pooling, followed by three combinations of a Dense-Block and Trans-Block, plus a final Dense-Block. Each Dense-Block consisted of two layers of convolution and batch normalization, whereas each Trans-Block consisted of three layers of batch normalization, convolution, and average pooling. The back-end comprised two Bidirectional Gated Recurrent Units (BGRUs) with a 100-class softmax layer for each of the 100 words in the LRW-1000 dataset. The accuracy of the D3D model was 34.76%.

In an interesting approach, Petridis et al. [20] presented an end-to-end audiovisual model based on residual networks and BGRUs. This model automatically extracted features from image and sound streams and performed word recognition. Martinez et al. [21] developed a word-based lip-reading system comparable to that of Petridis et al. [20] with a similar front-end consisting of a spatiotemporal CNN and a ResNet-18 CNN. For the back-end, the BGRU had been replaced with a network called the Multi-Scale Temporal Convolutional Network (MS-TCN), designed to customize the TCN's receptive field so that long- and short-term information could be combined. An MSTCN block comprised a series of TCNs, each with a unique kernel size, with the outputs concatenated. Their system was trained and evaluated using the English dataset LRW and the Mandarin dataset LRW-1000, achieving 85.3% and 41.5% word accuracy, respectively. In addition to increasing the system's accuracy, Petridis et al. [20] also observed a two-thirds reduction in GPU training time.

More recently, Ma et al. [22] proposed a modification of Martinez et al.'s system by employing a Densely Connected Temporal Convolutional Network (DC-TCN) instead of the MS-TCN present in the front-end in order to provide denser and more robust temporal features. Fully Dense (FD) and Partially Dense (PD) architectures were utilized, as well as an additional "Squeeze and Excitation" block within the network, which was a lightweight attention mechanism that further improved the classification power of the model. They achieved word accuracy rates of 88.4% and 43.7% for the LRW and LRW-1000 datasets, respectively, which was an improvement over Martinez et al. [21]. Notably, a speech training system was developed for individuals with hearing impairment and dysphonia using a CNN and an RNN [23]. First and foremost, a database for speech training was created to store the mouth shapes and gesture language vocabulary of individuals without impaired hearing or dysphonia. The system as a whole combined MobileNet and LSTM networks to conduct lip reading and then compared the result to the lip shapes of those with impaired hearing. Lastly, the system compared and analyzed the lip size, opening angle, lip shape, and other information of the individuals with impaired hearing and provided a standard lip-reading sequence for the individuals' learning and training.

Other interesting approaches based on state-of-the-art concepts in deep learning employ Gated Recurrent Units (a different form of RNN) to construct an encoder–decoder to learn phrases [24]. The attention mechanism was employed in the RNN for lip-reading recognition in combination with a CNN for image feature extraction [25]. The attention mechanism was also employed on visual data for speech representation using sub-word units [26]. Generative adversarial networks (GANs) were the basis of the visual context attention GAN (VCA-GAN) model [27], which integrates local and global lip movements for speech generation. Finally, graph convolutional networks were used for lip reading in the Adaptive Semantic-Spatio-Temporal Graph Convolution Network (ASST-GCN) model [28], which exploits dynamic mouth contours from visual data. For further reading, surveys dedicated to deep learning in lip reading have been recently published, such as [2,29].

## 3. Materials and Methods

The *Let's talk!* methodology, as illustrated in Figure 1, encompasses the preliminary stages of pre-processing, wherein procedures such as frame extraction, facial landmark identification, and lip segmentation are applied to the initial video data. These pre-processing steps contribute to the creation of the *MobLip* dataset, serving as the fundamental building block for subsequent stages. The synthesis of the *MobLip* dataset is a critical element in the pipeline, providing a robust foundation for training personalized models. Leveraging the 3D CNN and LSTM algorithm, these personalized models are tailored to individual patients, fostering a unique and precise approach to lip reading. Moving forward in the pipeline, the integration of these personalized models into the *Let's talk!* mobile application marks a seamless transition, ensuring accessibility and usability for end-users. The mobile application is engineered to harness the real-time frame capture capability inherent in smartphones, particularly utilizing the front camera for optimal user engagement. The activation

of this innovative feature initiates the integration of personalized models, thereby empowering users with access to a highly tailored and efficient speech recognition functionality. The strategic interplay of pre-processing, dataset synthesis, model training, and mobile application integration underscores the holistic and carefully orchestrated nature of the *Let's talk!* methodology.

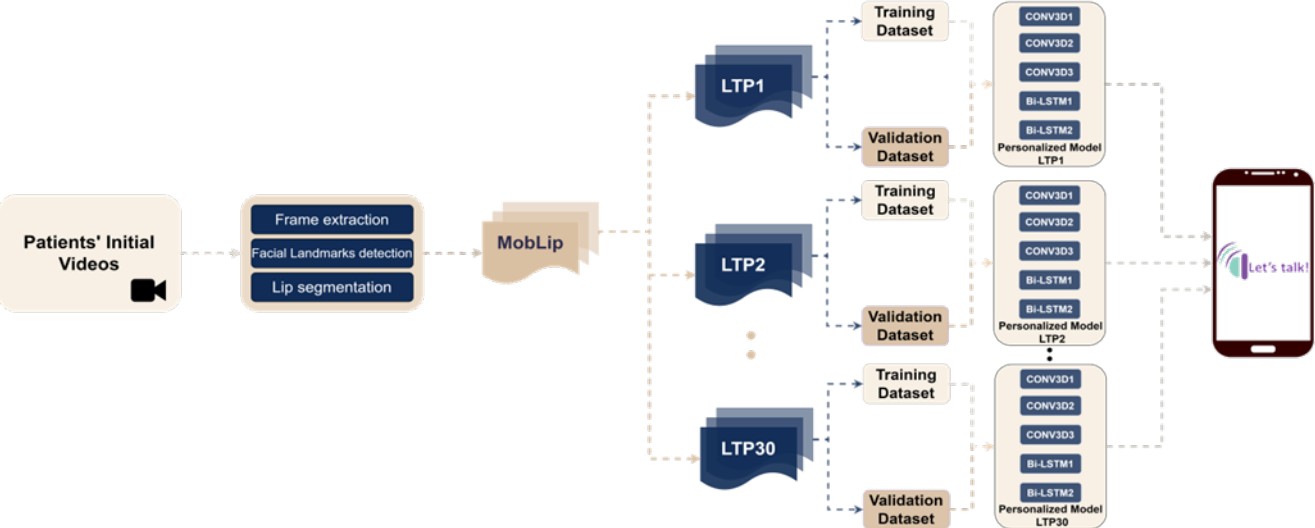

**Figure 1.** (**a**) The analysis and pre-processing of the video recordings, (**b**) *MobLip* dataset creation, and the integration of personalized models into the mobile application.

### 3.1. MobLip Dataset

The absence of publicly available datasets in the Greek language, along with the need to further evaluate the developed models, led to the collection of a new dataset, named *MobLip*. As part of the *Let's talk!* project, thirty participants were requested to recite a unique one-minute text, and their performance was captured on video. As for the sampling rate of the original video recordings, it was set to 30 frames per second. This frequency was chosen to provide high image quality to aid in the analysis of these data, which is critical for reliable data analysis and the subsequent training of the model. The selection

of the text recited by participants was a strategic process, tailored to encompass the most prevalent and representative words within the Greek language. The annotation process, an essential stage in dataset preparation, relied on a structured file accompanying each video recording. This file cataloged each word articulated in the recordings, precisely noting their corresponding start and end times. Notably, these annotations were made by clinical experts who are well versed in linguistic and speech analysis.

An essential subsequent step in dataset preparation involved the precise mapping of word durations to the sequences of lip movement frames. For this reason, the start and end times were converted into indices of frames by multiplying each time by the number of frames per second (fps) and dividing by 10,000 to obtain the start and end frames. An intriguing observation surfaced during the annotation phase: certain tiny words, including pronouns and grammatical articles, exhibited fewer associated frames compared to their larger counterparts. However, maintaining a consistent frame count for each word throughout the training phase stands as a pivotal requisite for word prediction models. Addressing this challenge necessitated the implementation of oversampling or undersampling techniques. These methodologies were applied to the extracted frames encompassing the lip regions corresponding to individual words, ensuring the preservation of a fixed number of words within each sequence. This exhaustive preparation and curation process culminated in the assembly of an extensive repository of 55,275 images, each bearing intrinsic linguistic and facial movement data. The resulting *MobLip* dataset not only serves as a testament to data collection and annotation but also stands as a foundational resource poised to invigorate advancements in Greek-language-based AI models, particularly in speech recognition driven by lip movement analysis. Table 1 provides a comprehensive overview of the dataset's distinctive features.

**Table 1.** *MobLip* dataset description.

| Feature | Description |
| --- | --- |
| Name | MobLip |
| Language | Greek |
| Collection purpose | *Let's talk!* project |
| Participants | 30 |
| Recording specification | 30 frames per second |
| Text content | Unique one-minute text with prevalent Greek words |
| Annotation | Each word annotated with start and end times in video frames |
| Frame mapping | Times converted to frame indices for word categorization |
| Frame adjustments | Oversampling/undersampling for consistent word frame count |
| Total images gathered | 55,275 |
| Totalnumber of training words | 3685 |

*3.2. Facial Landmark Detection*

Lip segmentation begins with the extraction of images that make up the original video recording. After a predetermined number of images have been extracted based on the sampling rate of the recording, the facial landmark detection algorithm can be applied. Various methods for facial landmark detection have been described in the literature, but regression-based methods predominate. Through the Dlib library, which implements a facial landmark estimator, the facial landmarks of *MobLip* dataset participants were obtained. The primary characteristics of the facial landmark detection algorithm are the employment of a set of trained regression trees to correctly estimate the landmark locations of the face directly from a sparse subset of pixel intensities and the usage of a sparse subset of pixel intensities. It utilizes a cascade of regression classifiers, with each classifier predicting an updated vector of facial landmarks based on the Gradient Boosting (GB) classifier. The following methodology combines weak classifiers with a strong one, with the objective of learning the set of regression trees that minimizes the sum of squared errors and achieves satisfactory real-time operation results.

### 3.3. Computational Models

Personalized models were developed from the *MobLip* dataset for each participant so that the model could be trained on specific images and, consequently, words. Each participant in the *MobLip* dataset formulated unique text, resulting in a unique training set of images and words. Thus, the algorithm "learns" each individual's speech pattern, facial characteristics, and lip shape. The developed personalized models were trained to recognize a person's unique speech patterns and vocal movements, making them more effective and efficient at comprehending the specific speech pattern. In addition, they are more adaptable, which aids in overcoming difficulties associated with differences in pronunciation or speech issues in general. On this basis, 29 distinct personalized models were trained on each participant's data (one subject was excluded from the *MobLip* dataset because, in most images, his lips were closed due to his difficulty speaking).

Multiple architectures were investigated and developed in order to select the most effective and efficient one for the *MobLip* dataset. The table below displays all the architectures that were evaluated to determine the optimal one for word recognition.

The pursuit of the most effective architecture for word recognition led to an extensive exploration of multiple neural network architectures, as presented in Table 2. Initial experimentation involved leveraging pretrained CNNs tailored for lip reading, including VGG-16, ResNet50, Inception V3, and MobileNetV2, which are commonly used for image recognition tasks. These models underwent extensive testing, both with and without recurrent neural networks (RNNs). Subsequent refinements involved employing transfer learning methodologies and retraining these models specifically on the *MobLip* dataset. Furthermore, alternative architectures including two-dimensional (2D) and three-dimensional (3D) CNNs and RNNs were evaluated. The fusion of a 3D CNN with an LSTM network yielded the most adept and efficient architecture for lip-reading tasks.

**Table 2.** Types of examined architectures.

| Type | Architectures | Dataset |
|------|---------------|---------|
| Pretrained CNN | Inception V3, VGG-16, ResNet-50, MobileNetV2 | ImageNet and transfer learning to the *MobLip* |
| Pretrained CNN and RNN | Inception V3, VGG-16, ResNet-50, MobileNetV2, along with LSTM | ImageNet and transfer learning to the *MobLip* |
| RNN | LSTM | *MobLip* |
| CNN | 2D-CNN, 3D-CNN | *MobLip* |
| CNN and RNN | 3D CNN and LSTM | *MobLip* |

The lip-reading algorithm developed as part of the *Let's talk!* project is a combination of a 3D CNN and LSTM. The input data consist of a sequence of fifteen RGB $160 \times 80$ images. The *MobLip* dataset is the source of the images used as input for the algorithm. The architecture begins with a series of 3D CNNs that derive spatiotemporal characteristics from the input data. Batch normalization is performed after each level of convolution to enhance the stability of the training. Rectified Linear Unit Activation Functions (ReLUs) introduce nonlinearity and boost the model's capacity for representation. After activating the first convolution layer, the spatial-dropout normalization technique is employed to randomly drop features from each image to prevent overfitting. Max-pooling layers decrease the sampling of feature maps and the computational complexity of the model, thereby enhancing its performance.

To prepare for the iterative layers, the output of the convolutional layers is reshaped using the TimeDistributed wrapper and converted to a vector. The bidirectional LSTM layers are utilized to model the sequence's temporal dependencies. The model's ability to comprehend sequential patterns is enhanced by its ability to capture information from both past and future time steps due to its bidirectional nature. Dropout is applied after

each level of LSTM to further normalize the model. Through fully connected layers (FCLs), the output of the final LSTM layer is transformed into a vector. The final dense layer with a softmax activation function generates a probability distribution over the target classes, indicating the likelihood of correctly predicting each class.

The 3D convolutional layers employ different parameters to extract features, such as padding with a size of (1, 2, 2) to preserve spatial dimensions, a stride of (1, 2, 2) for the first convolutional layer and (1, 1, 1) for the subsequent ones, and kernel sizes of (3, 5, 5) for the first two convolutional layers and (3, 3, 3) for the third one. The weights of the convolutional layers are initialized using the `he_normal` method. Batch normalization is applied after each convolutional layer to normalize the activations. ReLU activation functions are used to introduce nonlinearity and enhance the model's representational power. Spatial dropout with a rate of 0.25 is employed after the first activation layer to randomly drop out features within each frame and prevent overfitting. Max pooling is performed with a pool size of (1, 2, 2) and strides of (1, 2, 2) after each dropout layer to downsample the feature maps. The TimeDistributed wrapper is applied to reshape the output of the last max-pooling layer to prepare it for the recurrent layers. The bidirectional LSTM layers are configured with 256 units, `return_sequences` set to True to propagate information along the time steps, and the Orthogonal kernel initializer. Dropout with a rate of 0.5 is applied after each LSTM layer to further regularize the model. A flatten layer is used to reshape the output of the second LSTM layer before passing it through fully connected layers. The final dense layer has the number of units equal to the number of target classes and is initialized using the `he_normal` method. The softmax activation function is applied to produce a probability distribution over the classes, indicating the likelihood of each class prediction. These parameter settings, combined with the specified architecture, allow the model to effectively extract spatial and temporal features from the input video sequences and model long-term dependencies using recurrent layers.

Table 3 presents the key parameters employed during the training of the lip-reading model. The input shape, initialized with the dimensions (15, 160, 80, 3), represents a sequence of 15 frames, with each frame consisting of an image of size 160 × 80 pixels in RGB format. The subsequent layers unfold the specifics, including padding, convolutional strides, kernel sizes, and dropout rates, capturing the details of the 3D CNN and LSTM architecture. Notably, the bidirectional LSTM layers introduce memory units of 512, fostering the model's capacity for capturing temporal dependencies. The final dense layer, activated through softmax, outputs predictions for a variable number of classes. Additional training configurations, such as batch size (32) and epochs (500), are also documented. The model architecture is summarized in Figure 2.

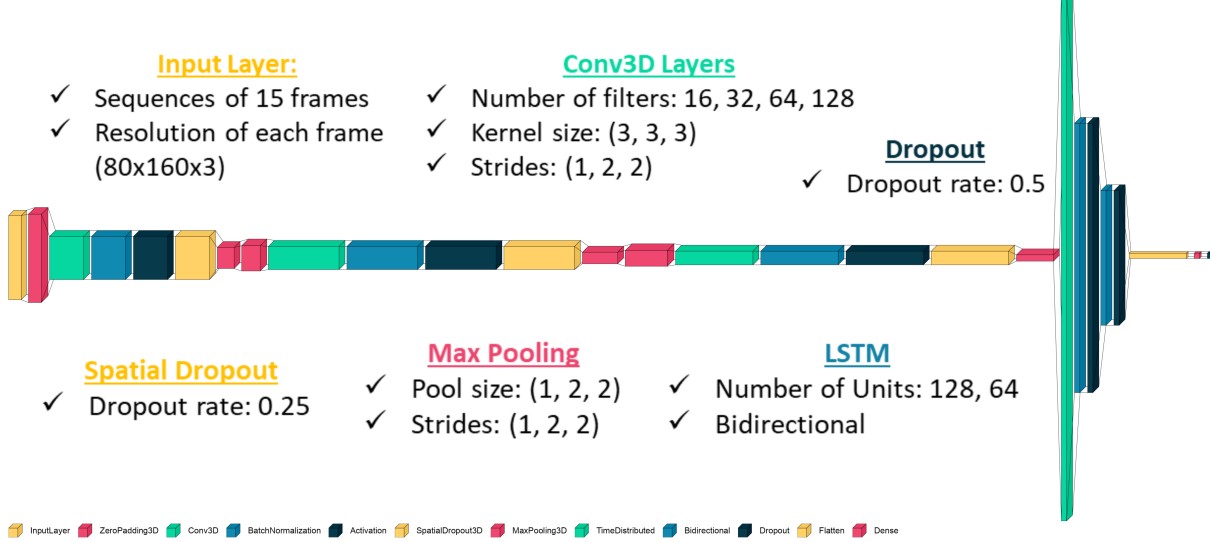

**Figure 2.** Let's Talk! architecture.

**Table 3.** 3D CNN and LSTM parameters.

| Layer | Parameter | Value | Output Shape |
|---|---|---|---|
| Input | - | - | (None, 15, 160, 80, 3) |
| ZeroPadding3D (zero1) | Padding | (1, 2, 2) | (None, 17, 164, 84, 3) |
| Conv3D (conv1) | Stride kernel size | (1, 2, 2), (3, 5, 5) | (None, 15, 80, 40, 32) |
| BatchNormalization (batc1) | - | - | (None, 15, 80, 40, 32) |
| Activation (actv1) | - | - | (None, 15, 80, 40, 32) |
| SpatialDropout3D | Rate | 0.25 | (None, 15, 80, 40, 32) |
| MaxPooling3D (max1) | Pool size | (1, 2, 2) | (None, 15, 40, 20, 32) |
| ZeroPadding3D (zero2) | Padding | (1, 2, 2) | (None, 17, 42, 22, 32) |
| Conv3D (conv2) | Stride kernel size | (1, 1, 1), (3, 3, 3) | (None, 15, 40, 20, 128) |
| BatchNormalization (batc2) | - | - | (None, 15, 40, 20, 128) |
| Activation (actv2) | - | - | (None, 15, 40, 20, 128) |
| SpatialDropout3D | Rate | 0.25 | (None, 15, 40, 20, 128) |
| MaxPooling3D (max2) | Pool Size | (1, 2, 2) | (None, 15, 20, 10, 128) |
| TimeDistributed | - | - | (None, 15, 25600) |
| Bidirectional (lstm1) | Units | 512 | (None, 15, 512) |
| Dropout | Rate | 0.5 | (None, 15, 512) |
| Bidirectional (lstm2) | Units | 512 | (None, 15, 512) |
| Dropout | Rate | 0.5 | (None, 15, 512) |
| Flatten | - | - | (None, 7680) |
| Dense (dense1) | Activation | Softmax | (None, number of classes) |
| Activation (softmax) | - | - | (None, number of classes) |
| - | Batch size | 32 | - |
| - | Epochs | 500 | - |

## 4. Results

While performing the annotation process, validation data were collected since each text contains words that appear more than twice at various times in the original video recording. These frames were not provided for training and were kept for the purpose of validating each personalized model. This method of verification is trustworthy because the previous and subsequent terms differ from those in the training set. In addition, the second time the participant pronounces the word, the manner of speech or pronunciation may be different, thereby validating the model's generalizability. The total number of images provided for validation is 4425. Table 4 shows the results from the *MobLip* participants obtained during the verification phase of the personalized models.

During the verification of the personalized models, the number of words spoken each time may have varied, depending on when the words were repeated in each video recording. As previously stated, each participant uttered a unique text, resulting in a variance in the number of words and, consequently, the number of resulting images. The results indicate that the model accurately anticipated the words for the majority of participants. The accuracy ranges from 28.57% to 87.50%, indicating that the efficacy of the various models varies. Greater percentages, such as 76.47%, 78.57%, and 87.50%, indicate relatively successful lip-reading abilities. Some models exhibit consistent performance, with percentages such as 75.00% and 62.50% indicating consistent results across a variety of people. The model with the lowest accuracy rate, 28.57%, received more words for validation than models with higher rates. Also, fewer words were provided for training than with the other models. Consequently, the accuracy rate is below average. Finally, it is important to note that the model developed for Let's Talk Participant 10 (LTP10) attained the highest rate of accuracy, 87.50%, among all personalized models.

In terms of the experimental setup, the training phase for a personalized model, specifically for LTP22, was completed efficiently, with a total time of 7 min and 15 s. This duration encompasses the entire training process, where our proposed model learned to recognize and classify lip movements associated with speech for LTP22. This rapid training time underscores the effectiveness of the model's architecture and optimization strategies, making it suitable for real-time or time-sensitive applications. The efficient

convergence achieved during training is indicative of the model's capability to swiftly adapt and learn from the provided data, showcasing its suitability for practical deployment in speech recognition systems based on lip movements.

**Table 4.** Results of the Let's Talk! personalized models.

| Participant | Training Phase | | Validation Phase | | Accuracy |
| --- | --- | --- | --- | --- | --- |
| | Words | Frames | Words | Frames | |
| LTP1 | 41 | 615 | 8 | 120 | 75.00% |
| LTP2 | 94 | 1410 | 13 | 195 | 62.50% |
| LTP3 | 114 | 1710 | 8 | 120 | 37.50% |
| LTP5 | 73 | 1095 | 8 | 120 | 45.45% |
| LTP6 | 80 | 1200 | 8 | 120 | 50.00% |
| LTP7 | 48 | 720 | 8 | 120 | 28.57% |
| LTP8 | 123 | 1845 | 8 | 120 | 76.47% |
| LTP9 | 98 | 1470 | 8 | 120 | 75.00% |
| LTP10 | 111 | 1665 | 8 | 120 | 87.50% |
| LTP11 | 121 | 1815 | 59 | 885 | 62.71% |
| LTP12 | 130 | 1950 | 8 | 120 | 50.00% |
| LTP13 | 131 | 1965 | 6 | 90 | 33.33% |
| LTP14 | 121 | 1815 | 8 | 120 | 75.00% |
| LTP15 | 119 | 1785 | 11 | 165 | 45.45% |
| LTP16 | 97 | 1455 | 6 | 90 | 33.33% |
| LTP17 | 178 | 2670 | 7 | 105 | 42.86% |
| LTP18 | 200 | 3000 | 9 | 135 | 55.56% |
| LTP19 | 135 | 2025 | 8 | 120 | 62.50% |
| LTP20 | 126 | 1890 | 8 | 120 | 62.50% |
| LTP21 | 191 | 2865 | 10 | 150 | 60.00% |
| LTP22 | 128 | 1920 | 14 | 210 | 78.57% |
| LTP23 | 167 | 2505 | 8 | 120 | 62.50% |
| LTP24 | 160 | 2400 | 8 | 120 | 60.00% |
| LTP25 | 135 | 2025 | 8 | 120 | 55.56% |
| LTP26 | 171 | 2565 | 8 | 120 | 75.00% |
| LTP27 | 144 | 2160 | 8 | 120 | 42.86% |
| LTP28 | 160 | 2400 | 8 | 120 | 62.50% |
| LTP29 | 117 | 1755 | 8 | 120 | 60.00% |
| LTP30 | 172 | 2580 | 8 | 120 | 37.50% |
| **TOTAL** | | **55,275** | | **4425** | |

The development of a generalized model comprising lip-reading data from multiple participants can also provide numerous benefits. By training a model on a diverse dataset comprising a large number of participants, the model can learn to generalize lip movements and speech patterns to various individuals. This enables the model to perceive and comprehend a broader range of lip movements and speech variations, thereby making it more robust and effective in a variety of real-world scenarios. Personalized models may be more scalable than models trained on data from a large population of participants. Rather than creating distinct models for each individual, a single generalized model can be created to serve a larger population. This reduces the need for extensive training and development for each individual and increases the accessibility of lip-reading technology. In light of this, a generalized model was developed from the *MobLip* dataset. Specifically, common terms spoken by these 30 participants were identified. After identifying these prevalent words, the most significant words encountered most frequently in daily life remained. This produced a subset of the *MobLip* dataset consisting of 41 words and 12.915 images, with each word being spoken by 20 unique participants. This data subset was used to train the 3D CNN and LSTM algorithm. The repeated terms were used for model verification, which was again based on the accuracy metric. The number of words provided for substantiation is eight, resulting in 2.070 images, with each word being spoken by at least 15 participants.

The accuracy rate is 60.00%, which indicates that the model correctly predicted nearly five of the eight words.

The results indicate that the personalized models achieved a higher rate of accuracy than the generalized model, as the algorithm learns each participant's speech pattern, facial features and lip shape. The developed personalized models were specifically trained to discern the unique speech patterns and lip movements of an individual, enabling them to comprehend the specific speech pattern more effectively and efficiently. In addition, they are more adaptable, which aids in overcoming difficulties associated with differences in pronunciation or speech disorders in general. Consequently, the *Let's talk!* methodology relies on personalized participant models.

## 5. Discussion

The results of the experiments showcase the effectiveness of the proposed personalized *Let's talk!* architecture in the lip-reading task. While most of the models achieved impressive accuracy rates, it is important to dissect these performance metrics further to gain a nuanced understanding of its capabilities and limitations. One notable aspect of the model's performance is its ability to handle noisy and uncontrolled environments. The robustness exhibited to variations in lighting conditions and speech patterns suggests that our approach is well-suited for real-world applications. However, it is essential to acknowledge that there may still be challenges in scenarios with unconventional speaking styles. In addition, the success of the proposed approach owes a significant debt to the quality and diversity of the dataset utilized for training and evaluation. The collected dataset, named *MobLip*, containing lip frames from 30 participants for recognizing specific Greek words, played a vital role in model's generalization capability. With an accuracy rate of 87.5%, the LTP10 participant's personalized model obtained the highest rate. Finally, the personalized models have been easily integrated into the *Let's talk!* mobile application through its HDF5 export format, and they can infer specific words from lip movements.

While the proposed investigation into lip-reading and speech recognition from facial movements has provided promising results, it is important to acknowledge certain limitations that may impact the scope and generalizability of the findings. Firstly , the *MobLip* dataset, although carefully curated for this study, may have inherent biases or lack a complete representation of all possible facial movements and speech patterns. Additionally, the performance of the lip-reading models was influenced by factors such as lighting conditions, facial variations, and individual speaking styles present in the dataset. Another crucial aspect to consider is the relatively limited number of participants included in the *MobLip* dataset. The restricted sample size may impact the generalizability of the findings and warrants caution when extrapolating the results to a broader population. Moreover, the computational demands associated with training deep learning models, especially those incorporating 3D convolutional and LSTM layers, can be substantial. Another consideration is the inherent complexity of the deep learning models that may present challenges in understanding the decision-making process. Furthermore, the specific focus on the Greek language in the dataset may limit the applicability of the lip-reading models to other languages, and additional research is needed to explore cross-language performance. By explicitly acknowledging these limitations, the proposed solution aims to provide a transparent view of the boundaries and potential constraints of the lip-reading task, encouraging future work to address and overcome these challenges for further advancements in the field.

In conclusion, we introduced a novel approach for the recognition of spoken words from lip movements using a combination of 3D CNNs and LSTM networks with an accuracy rate of up to 87.5%. Our research aimed to address the challenges associated with lip reading in noisy and real-world environments, with potential applications in assistive technologies, human–computer interaction, and security systems. *Let's talk!* offers the first deep learning neural network architecture that is successfully implemented for identifying Greek words from diverse time frames with low loss and high accuracy rates.

The architecture adheres to a similar philosophy to that of LipNet [16], the state-of-the-art implementation for speech recognition, without being constrained by a specific participant's speech pattern or the number of neurons. In addition, the proposed architecture was trained on lip images corresponding to video recordings of participants containing a vast array of Greek phonemes.

In future work, the word prediction model will be expanded to provide word predictions not only in text format but also in audio format, as well as provide notifications regarding the use of the application and the execution time of the model for the connected user. These word predictions will be appropriately converted into a personalized audio format, where the distinctive voice of each participant will be incorporated into the audio information's frequency spectrum to provide a more realistic effect of the word predictions via the mobile application. Lastly, the overarching objective is to develop a word prediction model that can be used for lip reading in the general population.

**Author Contributions:** Conceptualization, T.E.; methodology, T.E.; software T.E.; validation, G.C., Z.Z. and E.K.; formal analysis, T.E., G.N.D. and A.G.V.; investigation, T.E., G.C., Z.Z. and E.K.; resources, G.C., Z.Z. and E.K.; data curation, G.C., Z.Z. and E.K.; writing—original draft preparation, T.E., G.N.D. and A.G.V.; writing—review and editing, T.E., G.N.D. and A.G.V.; visualization, G.N.D. and A.G.V.; supervision, T.E.; project administration, T.E.; funding acquisition, T.E. All authors have read and agreed to the published version of the manuscript.

**Funding:** This research work was supported by the Hellenic Foundation for Research and Innovation (H.F.R.I.) under the "First Call for H.F.R.I. Research Projects to support Faculty members and Researchers and the procurement of high-cost research equipment grant" (Project Number: HFRI-FM17-579) and "Let's Talk!: Application of multimodal interfaces for people with loss of voice forspeech reproduction with natural voice".

**Institutional Review Board Statement:** The study was conducted in accordance with the Declaration of Helsinki and approved by the Ethics Committee of Ippokration Hospital of Athens (Protocol Number 8/42, 18 November 2020).

**Informed Consent Statement:** Informed consent was obtained from all subjects involved in the study.

**Data Availability Statement:** Dataset available on request from the authors.

**Conflicts of Interest:** The authors declare no conflicts of interest. The funders had no role in the design of the study; in the collection, analyses, or interpretation of data; in the writing of the manuscript; or in the decision to publish the results.

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
