# Peer review of "Lip-Reading Advancements: A 3D Convolutional Neural Network/Long Short-Term Memory Fusion for Precise Word Recognition"

_biomedinformatics, doi:10.3390/biomedinformatics4010023_

Round 1

Reviewer 1 Report

Comments and Suggestions for Authors

1.     Describe dataset features in more details and its total size and size of (train/test) as a table.

2.     Pseudocode / Flowchart and algorithm steps need to be inserted.

3.     Time spent need to be measured in the experimental results.

4.     Limitation and Discussion Sections need to be inserted.

5.     All metrics need to be calculated such as accuracy, Precision, Recall, Loss, F1 score, ROC AUC score and Confusion matrix in the experimental results.

6.     The parameters used for the analysis must be provided in table

7.     The architecture of the proposed model must be provided

8.     Address the accuracy/improvement percentages in the abstract and in the conclusion sections, as well as the significance of these results.

9.     The authors need to make a clear proofread to avoid grammatical mistakes and typo errors.

10.   Add future work in last section (conclusion) (if any)

11.   To improve the Related Work and Introduction sections authors are recommended to review this highly related research work paper:

a)     A Novel Hybrid Approach to Masked Face Recognition using Robust PCA and GOA Optimizer

b)    Optimizing epileptic seizure recognition performance with feature scaling and dropout layers

Author Response

Comments and Suggestions from First Reviewer

  1. Describe dataset features in more details and its total size and size of (train/test) as a table.

Answer: The MobLip dataset has been elucidated in the revised manuscript, with an in-depth exposition provided in Table 1. This table furnishes a comprehensive overview of the dataset's distinctive features, encompassing pertinent details such as the dataset's name, language, the purpose of its collection, the number of participants, recording specifications, textual content specifics, annotations, frame mapping techniques, frame adjustments for consistent word frame count, and the unique identifier assigned to each participant. Additionally, critical statistics such as the total number of images gathered have been inclusively incorporated into the table.

  1.  Pseudocode / Flowchart and algorithm steps need to be inserted.

Answer: The synthesized MobLip dataset, along with the subsequent stages encompassing the training of personalized models and the ultimate integration into the mobile application, is comprehensively depicted through a meticulously crafted flowchart, as presented in Figure 2. This graphical representation depicts the sequential steps undertaken in the dataset synthesis process and the subsequent model training phases.

  1. Time spent need to be measured in the experimental results.

Answer: The temporal investment for the experimental results pertaining to Let's Talk Patient 22 (LTP22) is explicitly detailed within the Results section of the manuscript. This information offers transparency regarding the duration allocated to the experimentation process for this specific patient, contributing to a comprehensive understanding of the temporal aspects associated with the study's outcomes.

  1. Limitation and Discussion Sections need to be inserted.

Answer: The Limitation and Discussion sections have been incorporated into the manuscript to address critical aspects of the study. In the Limitation section, particular attention is given to the restricted number of patients, acknowledging its significance as a fundamental limitation of the study. This disclosure aims to provide a clear overview of the constraints and potential boundaries inherent in the study. Subsequently, the Discussion section delves into an in-depth analysis of the study's findings, emphasizing key insights and implications.

  1. All metrics need to be calculated such as accuracy, Precision, Recall, Loss, F1 score, ROC AUC score and Confusion matrix in the experimental results.

Answer: The accuracy metric reflects the correctly identified number of words and it is related to the word error rate (WER) metric which has been previously reported in the literature[1]. In this study we are trying to solve a multiclass classification problem, where the number of samples for testing is small. To this end, we are not able to provide measures such as specificity, sensitivity, AUC, and F1 score since they reflect the per-class measurement of performance in terms of true positives false positives, true negatives, and false negatives. Regarding the confusion matrix, as presented in Table 4, the number of words during the training and validation phase are too high so the matrix will be huge.

  1. The parameters used for the analysis must be provided in table

Answer: The manuscript has been updated to incorporate Table 3, which offers comprehensive insights into the parameters employed during the model training process.

  1. The architecture of the proposed model must be provided

Answer: The architecture of the proposed model is elucidated in Figure 1, complemented by a extensive explication in Section 2, titled "Methodology." To further enhance the clarity and transparency of the proposed methodology, Table 3 has been incorporated, delineating the parameters employed in the construction of the model.

  1. Address the accuracy/improvement percentages in the abstract and in the conclusion sections, as well as the significance of these results.

Answer: In the revised abstract and conclusion sections, percentages have been incorporated to enhance precision and clarity.

  1. The authors need to make a clear proofread to avoid grammatical mistakes and typo errors.

Answer: The manuscript has undergone an extensive proofreading process to eliminate grammatical errors and typographical mistakes, ensuring enhanced clarity and readability in the content.

  1. Add future work in last section (conclusion) (if any)

Answer: The conclusions section has been augmented with considerations for future work, providing valuable insights into potential avenues for further research and development in this domain.

  1. To improve the Related Work and Introduction sections authors are recommended to review this highly related research work paper: a) A Novel Hybrid Approach to Masked Face Recognition using Robust PCA and GOA Optimizer b) Optimizing epileptic seizure recognition performance with feature scaling and dropout layers

Answer: The proposed studies have been incorporated into the introduction section.

[1] Assael, Y. M., Shillingford, B., Whiteson, S., & De Freitas, N. (2016). Lipnet: End-to-end sentence-level lipreading. arXiv preprint arXiv:1611.01599.

Reviewer 2 Report

Comments and Suggestions for Authors

The authors propose a model that fuses 3D-CNN and LSTM to recognize lips in videos. This seems like an interesting idea, but I have some major concerns.

 1.       The author's sentences are too long, with several of them being more than 40 words long. Such length leads to poor readability. For example, sentences in lines 16-19, 23-26, etc. It is recommended that more short sentences be used to improve readability.

2.       The authors propose a fusion model of 3D-CNN and LSTM. What is the motivation? Please provide a full description in the paper. As far as I know 3D-CNN can also learn the temporal order relationship between each image, what is the motivation for using LSTM?

3.       Nowadays, various advanced models of CNN are widely used in biomedical image processing. The authors should discuss these advanced models in detail in the introduction. For example:https://ieeexplore.ieee.org/abstract/document/9875035, https://ieeexplore.ieee.org/document/9527115, https://www.sciencedirect.com/science/article/abs/pii/S0010482522011556.

4.       Second, the Transformer model has become mainstream in the field of biomedical image recognition. The authors should discuss the advantages and disadvantages of this method over the Transformer method in detail in the introduction. For example:

https://ieeexplore.ieee.org/document/9868801,

https://link.springer.com/article/10.1007/s00432-023-04795-y,

https://www.oejournal.org/article/doi/10.12086/oee.2023.220158.

5. Where is the full name of LTP? I don't seem to see it, please define the acronym strictly.

6. The authors did not describe the data set in detail. Please describe the data set in a list.

7. The authors do not provide details on the model assessment indicators. Please add details.

8. where are the ablation experiments for 3D-CNN and LSTM?

9. The author's overall charts and graphs are simply too few. It doesn't seem like an academic paper. Please add more experimental and analytical graphs.

Comments on the Quality of English Language

None

Author Response

Comments and Suggestions from Second Reviewer

  1. The author's sentences are too long, with several of them being more than 40 words long. Such length leads to poor readability. For example, sentences in lines 16-19, 23-26, etc. It is recommended that more short sentences be used to improve readability.

Answer: The paper underwent a process of rephrasing, incorporating succinct sentences to enhance clarity and readability.

  1. The authors propose a fusion model of 3D-CNN and LSTM. What is the motivation? Please provide a full description in the paper. As far as I know 3D-CNN can also learn the temporal order relationship between each image, what is the motivation for using LSTM?

Answer: The bidirectional architecture of the LSTM layers significantly contributes to the model's capacity to capture temporal dependencies spanning both past and future time steps, a critical aspect for effective sequential data analysis. Furthermore, experimental results indicate that the integration of Convolutional Neural Networks (CNN) with Recurrent Neural Networks (RNN) resulted in superior performance, particularly evident in the accuracy metric.

  1. Nowadays, various advanced models of CNN are widely used in biomedical image processing. The authors should discuss these advanced models in detail in the introduction. For example: https://ieeexplore.ieee.org/abstract/document/9875035, https://ieeexplore.ieee.org/document/9527115, https://www.sciencedirect.com/science/article/abs/pii/S0010482522011556.

Answer: The proposed studies have been incorporated into the introduction section.

  1. Second, the Transformer model has become mainstream in the field of biomedical image recognition. The authors should discuss the advantages and disadvantages of this method over the Transformer method in detail in the introduction. For example:

https://ieeexplore.ieee.org/document/9868801,

https://link.springer.com/article/10.1007/s00432-023-04795-y,

https://www.oejournal.org/article/doi/10.12086/oee.2023.220158.

Answer: The proposed studies have been incorporated into the introduction section.

  1. Where is the full name of LTP? I don't seem to see it, please define the acronym strictly.

Answer: The acronym "LTP" (Let's Talk Patient) is explicitly mentioned in Table 1 under the "Patient ID" column. Also, it is mentioned in the paper with the abbreviation.

  1. The authors did not describe the data set in detail. Please describe the data set in a list.

Answer: The MobLip dataset has been elucidated in the revised manuscript, with an in-depth exposition provided in Table 1. This table furnishes a comprehensive overview of the dataset's distinctive features, encompassing pertinent details such as the dataset's name, language, the purpose of its collection, the number of participants, recording specifications, textual content specifics, annotations, frame mapping techniques, frame adjustments for consistent word frame count, and the unique identifier assigned to each participant. Additionally, critical statistics such as the total number of images gathered have been inclusively incorporated into the table.

  1. The authors do not provide details on the model assessment indicators. Please add details.

Answer:  The accuracy metric reflects the correctly identified number of words and it is related to the word error rate (WER) metric which has been previously reported in the literature[1]. In this study we are trying to solve a multiclass classification problem, where the number of samples for testing is small. To this end, we are not able to provide measures such as specificity, sensitivity, AUC, and F1 score since they reflect the per-class measurement of performance in terms of true positives false positives, true negatives, and false negatives. External validation except the train-test splitting of the dataset is also mentioned in the Results section.

  1. Where are the ablation experiments for 3D-CNN and LSTM?

Answer: The architectures that underwent testing to determine the most accurate configuration within the Let's Talk methodology are comprehensively presented in Table 2.

  1. The author's overall charts and graphs are simply too few. It doesn't seem like an academic paper. Please add more experimental and analytical graphs.

Answer:  A new graph has been incorporated (Figure 2) describing the system architecture.

[1] Assael, Y. M., Shillingford, B., Whiteson, S., & De Freitas, N. (2016). Lipnet: End-to-end sentence-level lipreading. arXiv preprint arXiv:1611.01599.

Reviewer 3 Report

Comments and Suggestions for Authors

The paper addresses a significant subject: Lip-reading. The authors propose an amalgamation of 3D Convolutional Neural Networks (CNNs) and Long Short-Term Memory (LSTM) networks to enhance word recognition from lip movements specifically using the Greek language. The paper's structure is well organized, the methodology is lucid, and the results are intriguing.

To align with the journal's scope, it is necessary for the authors to expand the introduction by exploring potential applications of Lip-Reading in medicine or health more broadly.

In addition, it would be beneficial for the authors to elaborate on the creation process of the dataset, potentially by providing excerpts or examples of the texts used to compile it. Furthermore, for the sake of reproducibility, including a link to the dataset would greatly enhance the paper's accessibility.

Author Response

Comments and Suggestions from Third Reviewer

  1. To align with the journal's scope, it is necessary for the authors to expand the introduction by exploring potential applications of Lip-Reading in medicine or health more broadly.

Answer: An additional paragraph has been introduced in the introduction section, delving into the exploration of potential applications of Lip-Reading in the field of medicine or health more broadly.

  1. In addition, it would be beneficial for the authors to elaborate on the creation process of the dataset, potentially by providing excerpts or examples of the texts used to compile it. Furthermore, for the sake of reproducibility, including a link to the dataset would greatly enhance the paper's accessibility.

Answer: In adherence to ethical considerations, the dataset utilized in this study is not publicly available, and as such, a link to access the dataset cannot be provided.

Round 2

Reviewer 1 Report

Comments and Suggestions for Authors

Accept.

Author Response

We would like to thank the reviewer

Reviewer 2 Report

Comments and Suggestions for Authors

The authors have addressed my concerns well.

Author Response

We would like to thank the reviewer

Reviewer 3 Report

Comments and Suggestions for Authors

The authors significantly improved the quality of the paper. No other comments.

Author Response

We would like to thank the reviewer